# Snapshots of ABCG1 and ABCG5/G8: A Sterol’s Journey to Cross the Cellular Membranes

**DOI:** 10.3390/ijms24010484

**Published:** 2022-12-28

**Authors:** Fatemeh Rezaei, Danny Farhat, Gonca Gursu, Sabrina Samnani, Jyh-Yeuan Lee

**Affiliations:** 1Department of Biochemistry, Microbiology, and Immunology, Faculty of Medicine, University of Ottawa, Ottawa, ON K1H 8M5, Canada; 2Biochemistry Program, Faculty of Science, University of Ottawa, Ottawa, ON K1H 6N5, Canada

**Keywords:** ABCG1, ABCG5, ABCG8, structural motif, sterol trasportation, TICE, RCT

## Abstract

The subfamily-G ATP-binding cassette (ABCG) transporters play important roles in regulating cholesterol homeostasis. Recent progress in the structural data of ABCG1 and ABCG5/G8 disclose putative sterol binding sites that suggest the possible cholesterol translocation pathway. ABCG1 and ABCG5/G8 share high similarity in the overall molecular architecture, and both transporters appear to use several unique structural motifs to facilitate cholesterol transport along this pathway, including the phenylalanine highway and the hydrophobic valve. Interestingly, ABCG5/G8 is known to transport cholesterol and phytosterols, whereas ABCG1 seems to exclusively transport cholesterol. Ligand docking analysis indeed suggests a difference in recruiting sterol molecules to the known sterol-binding sites. Here, we further discuss how the different and shared structural features are relevant to their physiological functions, and finally provide our perspective on future studies in ABCG cholesterol transporters.

## 1. Introduction

ATP-binding cassette (ABC) transporters are ubiquitous in all living organisms, and play a crucial role in maintaining body homeostasis by functioning as pumps. They actively transport a variety of substrates across cell membranes utilizing ATP catalysis [1,2,3], the mechanism that is congruent with the ATPase function at their cytoplasmic nucleotide binding domain (NBD). The transmembrane domain (TMD) collaborates with the NBD to initiate the transport of substrates while anchoring the protein within the bilayer [3]. On the basis of similarities in the gene and amino acid sequences, as well as the structural topology of human ABC proteins, they are classified into seven subfamilies, ranging from ABCA to ABCG [4,5].

So far, mammalian ABCG proteins are uniquely known to efflux lipid-like compounds, such as cholesterol, and recent progress in ABCG structural data has underscored their importance in understanding the mechanism of ABC transporters [6]. Cholesterol is a key component of cell membranes and the precursor of bile acids, steroid hormones and vitamin D which are vital for body health [7]. Cholesterol can be obtained by de novo biosynthesis or via dietary uptakes in the guts. However, not all imported cholesterol is metabolized in the cells, and the excess amount of free cholesterol has to be cleared via two metabolic pathways, namely trans-intestinal cholesterol efflux (TICE) and reverse cholesterol transport (RCT) [8,9]. ABCG cholesterol transporters, i.e., ABCG1, ABCG4, ABCG5, and ABCG8 contribute to different stages of these removal processes [10]. Pathogenic mutations in these transporters have been known to be associated with various metabolic and cardiovascular diseases [11,12].

ABCG1 is a homodimer localized to the plasma membrane and late endosomes, functioning synergistically with ABCA1 to facilitate cholesterol efflux toward high-density lipoprotein (HDL) particles and thereby reducing total cholesterol in macrophages and peripheral tissues [13,14]. Alongside cholesterol, ABCG1 also transports phospholipids, though the specifics of the protein-ligand interactions are still poorly understood [15]. The malfunction of ABCG1 and ABCA1 results in Tangier disease, a severe form of familial HDL deficiency [16]. 

The ABCG5/G8 transporter is an obligatory heterodimer predominantly expressed on the apical surface of hepatocytes along the bile ducts and of enterocytes on the brush-boarder membranes [17]. ABCG5/G8 is the primary transporter in hepatobiliary and trans-intestinal cholesterol secretion. This protein was first described in the disease Sitosterolemia, a rare hereditary lipid storage disorder, characterized by the buildup of plant sterols (i.e., phytosterols) in plasma and tissues [18,19]. In addition to the phytosterols, sitosterolemic patients absorb a greater portion of dietary cholesterol, and they release less cholesterol into the bile. If lacking normal de novo cholesterol synthesis by HMG-CoA reductase activity, sitosterolemic patients can suffer much higher absorption of plant sterols through the intestine [11,20,21]. 

Due to their importance in regulating cholesterol homeostasis, ABCG sterol transporters have received a lot of attention in physiological studies and therapeutic development; yet, there remains a growing need to investigate their structure-function relationship to understand their molecular mechanism. Recent progress in the determination of ABCG1 and ABCG5/G8 structures has shed light on the biochemical and functional aspects of ABCG sterol proteins. In this article, we will highlight the available structural data of ABCG1 and ABCG5/G8 and the resolved cholesterol binding sites, and propose reasoning for the molecular mechanism applied by these transporters.

## 2. The General Structure of ABCG1 and ABCG5/G8 

Site-directed mutagenesis studies and the crystal structure of ABCG5/G8, the first ABCG protein to have its structure identified, gave researchers a general notion of the structure and function of ABCG proteins [22,23,24]. The recent elucidation of other ABCG protein structures has provided more details regarding the organization of structural motifs and elements in these transporters [25,26]. The general structure of each ABCG half-transporter consists of a TMD, NBD, and an extracellular domain (ECD) [22]. Homo- or heterodimerization is required for these proteins to be expressed on the cell surface [27]. In a functional state, these proteins have six transmembrane helices (TMHs) in each half-transporter. TMH2 and 5 are central within the dimer interface, likely creating a pathway for substrate translocation. The distances between TMH1 and 2 as well as TMH3 and 4 form two loops, designated extracellular loop1 (ECL1) and ECL2, which are part of the ECD. Before coming to an end with TMH6, the C terminus of TMH5 forms a well-conserved re-entry helix which stretches to ECL3 [22,28]. The re-entry helices and ECLs contain conserved residues and form ECDs which may cooperate potentially in substrate release and direct binding of acceptor molecules [29,30]. The N-terminal portion of the TMH1 contains an orthogonal connecting helix (CnH), which is interfacial to the membrane bilayer and serves as the link between the TMD and the NBD. In this area, there is a coupling helix (CpH) formed by the loop between TMH2 and TMH3. Additionally, a helix with a conserved glutamate identifies the E-helix is present in close proximity to CnH. The CnH, CpH, and E-helix form a triple-helical bundle in this region, providing an immediate interface between NBD and TMD [22].

## 3. A Closer Look at the Structures of ABCG1 and ABCG5/G8

### 3.1. ECD: Hydrophobic Valve, ECLs

Just beneath the ECLs, TMH5 contains two conserved hydrophobic residues in ABCG members, primarily phenylalanines in the sterol transporters (Figure 1A,C). Studies on drug-efflux ABCG2 have suggested that these residues function as a hydrophobic valve to control entry to the upper cavity, receiving substrate before releasing it into the extracellular side [22,29,31,32]. This process is assumed to occur when the valves close during an inward relaxed conformation and open upon ATP hydrolysis during the change to an outward conformation. In addition to the hydrophobic valve, the ECLs also contain several conserved residues in ABCG proteins and some disease-causing mutations have been located in this area showing the importance of the ECDs in the substrate efflux to the extracellular matrix [33]. Moreover, in ECL3, ABCG5 and ABCG8 have a total of five cysteine residues, whereas the ABCG1 monomer has two cysteine residues which are believed to form intramolecular disulfide bonds [34]. Interestingly, ABCG1 has a shorter ECL3 than ABCG5/G8 and does not contain an N-glycosylation site [35]. 

### 3.2. TMD: Polar Relay, Cholesterol-Sensing Motif, Phenylalanine Highway

In close proximity to CnH and CpH in ABCG5/G8, a network of conserved polar residues, primarily Histidine, Asparagine, Glutamic acid, and Lysine form a polar relay generated by a network of hydrogen bonds and salt bridges (Figure 1A). This region is located immediately above the triple-helical bundle and is believed to act as a flexible hinge for subunit motions with a low energy barrier. The hydrogen bonds tying the TMD polar relay together may make this network more malleable than a buried hydrophobic core [22]. In ABCG1, this relay has yet to be defined. Here, using multiple sequence alignment analysis in the same region as that in ABCG5/G8, we identified several conserved polar residues that could potentially perform a comparable role as a polar relay in ABCG1 (Figure 1B). 

The cholesterol recognition amino acid consensus (CRAC) motif, characterized by the sequence marker of L/V-X (1-5)-Y-X (1-5)-R/K, was initially described in enzymes that are involved in cholesterol production, such as HMG CoA Reductase and SCAP [36]. In ABCG members, this motif was first described in ABCG1 as one of the cholesterol sensing motifs. Former research revealed that tyrosine residues, Y649, Y667, and Y672 on TMH6 are components of this motif in ABCG1 [37]. This motif is critical for both protein function and stability and can be utilized to monitor the membrane cholesterol level by the protein. In previous research, the stability of the Y667L mutant protein in the presence and absence of cholesterol was studied. This revealed that unlike Y649L and Y672L, Y667L displays reduced cholesterol efflux, but can still be stabilized by the addition of cholesterol, suggesting that Y667 is essential for protein stability. A sequence analysis study showed that the tyrosine residues representing this motif are conserved in almost all ABCG proteins with the exception of ABCG5 [37,38,39] (Figure 1C). In ABCG8, this element is represented by three residues of Y644, Y659, and F664 (Figure 1A). The role of this motif in ABCG8 has not been determined yet, but based on the structural homology and sequence conservation, we hypothesize that it contributes to the sterol sensing function of ABCG5/G8 [40].

By analyzing the amino acid sequences of ABCG proteins, we recently found several well-conserved phenylalanine residues on TMH2 directly beneath the hydrophobic valve and above a cavity that is open to the cytoplasm (Figure 1A,C). The equivalent residues on ABCG5 are replaced by tyrosines, yet maintain the same spatial conservation towards the TMD dimer interface. These residues, particularly the first and the last phenylalanine, create an array of phenylalanine clusters in this region, namely the phenylalanine highway. We have proposed that this phenylalanine highway may play a role in ligand capture, through the molecular clamp, as well as the subsequent maintenance of substrate orientation and transfer to the hydrophobic valve [41].

### 3.3. NBD: Walker A, Walker B, Signature Motif, NPXDF Motif

The canonical Walker A, Walker B, and signature motifs are three highly conserved regions within the NBDs of ABCG transporters that are essential for nucleotide binding and subsequent hydrolysis [11,42] (Figure 1A,C). The interactions of one monomer’s Walker A and B motifs with the signature motif of the opposing monomer form NBDs of each transporter. ATP is believed to bind at the junction of two NBDs, where Walker A stabilizes this bond mostly through a conserved lysine or arginine residue and Walker B motif’s aspartate recruits a magnesium ion into the nucleotide-binding site [43]. The ATP catalysis depends on the conserved catalytic glutamate, located just downstream of the Walker B motif [44,45,46,47]. In ABCG sterol transporters, although no enzymatic study has been reported yet, it is hypothesized that the carboxylic group of glutamate coordinates a water molecule that promotes nucleophilic attack on the γ-phosphate of bound ATP to initiate its hydrolysis [43,47]. 

Previous studies found that biliary cholesterol secretion was blocked by mutations of critical residues in the Walker A and Walker B motifs of ABCG5, but not by mutations at comparable sites in G8. When mutations were placed in the signature motif, the opposite outcome was seen. These findings collectively showed that the ABCG5 and ABCG8 NBDs are not functionally symmetric [47]. In ABCG1, mutating an essential glycine residue to alanine in the Walker A motif prevented its localization to the cell surface, abolished ABCG1′s ability to promote cholesterol esterification by oxidase and ACAT, and downregulated HDL-driven cholesterol efflux on the cultured mammalian cells. This indicated that the ATP binding domain in ABCG1 is essential for both lipid transport activity and protein trafficking [48]. 

NPXDF is a unique NBD motif in ABCG proteins and was first highlighted by a functional study on ABCG1. In this study, it has been unveiled that the cells expressing mutant ABCG1 for this motif show impaired cholesterol efflux activity compared with wild-type-expressing cells [49]. This motif was found to be closed upon dimerization between the two NBDs of half transporters, based on the ABCG5/G8 crystal structure and the ABCG2 cryo-EM structure [41,50]. It has been shown that key residues from the NPXDF motifs including two aspartate residues at the C-terminus of the short helical motif, NPXDFXXD, form two salt bridges with a positively charged arginine residue from the opposing half transporter. In ABCG5/G8, the first salt bridge is formed between the side chain of ABCG5_Arg253_ and the side chains of two aspartic acid residues, ABCG8_Asp319, Asp323_. The second pair is formed between the side chain of ABCG8_Arg273_ and the side chains of two aspartic acid residues, ABCG5_Asp299, Asp303_ (Figure 1A). Sequence alignment of the NPXDFXXD motif suggests that the residues forming the salt bridges are conserved in the ABCG family and across species [50] (Figure 1C). 

## 4. Revelation of Cholesterol Binding Sites in ABCG Sterol Transporters and Logical Working Models

### 4.1. ABCG5/G8

We recently determined a crystal structure of ABCG5/G8 in a cholesterol-complexed state. This structure provided evidence supporting the existence of a cholesterol molecule binding on the ABCG5-dominant side in the interface of two symmetrical proteins (site 3) [41] (Figure 2A(a)). In this peripheral binding site, cholesterol makes direct contact with the conserved ABCG5_A540_ residue. A mutant ABCG5_A540F_ was previously shown to downregulate biliary cholesterol efflux in vivo and inhibited sterol-coupled ATPase activity in vitro [22]. The other binding site (site 1) within ABCG5/G8 has been determined to be located in the inner leaflet of the membrane (Figure 2A(b)). In this site, the isooctyl side chain of cholesterol facing the intracellular matrix of the membrane, close to Ile529 in TMH5 of ABCG5 and to Ile419 and Leu465 in TMH1 and TMH2 of ABCG8, respectively. The remaining bound cholesterol was found deeper within the central cavity of the TMD, situated halfway between the intracellular and extracellular leaflets and oriented parallel to the membrane on the ABCG8-dominant side (site 2) [25] (Figure 2A(c)). 

All three cholesterol binding sites on ABCG5/G8 take on uncommon “flipped” conformation in relation to the membrane’s general amphipathic nature (Figure 2A). Such flipped cholesterol conformations have been previously observed in membrane dynamics research as well as bound to other membrane proteins [51,52]. Spontaneous cholesterol flipping between leaflets is a common event in the plasma membrane, capable of undertaking intermediate horizontal orientation within the membrane’s core [51]. These orientations are similar to what is observed in ABCG5/G8′s binding sites 2 and 3. Site 1 contains a vertically oriented cholesterol molecule in a flipped conformation. The presence of hydrophobic residues as well as charged amino acids near the polar head likely helps bind and stabilize the sterol conformation [25]. The presence of this sterol also aids protein function as found through in vivo mouse biliary cholesterol transport [22]. Overhead near alanine 540 is site 3, which was found to be an important residue [22,40]. It is thus possible that ABCG5/G8 catalyzes cholesterol flipping from inner to outer leaflets, peripherally through its exterior surface, like P4-ATPase “credit card” models [53]. On the opposite dimer interface (site 2), a horizontal cholesterol molecule has been resolved and was able to bind cholesterol from a neutral orthogonal position within the membrane core, as seen through molecular dynamic simulations [25] (Figure 3A). However, the initial transition toward that horizontal state remains to be unknown, a future direction to further investigate the transporter’s mechanism.

ABCG5/G8 is presented on the canalicular membranes’ outer leaflet, enriched with cholesterol and sphingomyelin (SM), two lipids that can generate liquid-ordered phases of membranes, causing asymmetrical phases between the membrane [54,55,56]. Liquid-ordered phases, equally referred to as lipid rafts or lipid microdomains, represent dense and rigid structures within the fluid plasma membrane, exhibiting detergent resistance [39,57,58]. This type of resistance is crucial when localized so closely to the bile, containing waste product, cholesterol and bile salts. The canalicular membrane experiences stringent interactions with bile acids’ detergent properties, leading some to speculate about the self-preserving functions of high-density lipid packing in the outer leaflets of the plasma membranes on the apical cell surface [59,60]. As discovered by an earlier study, membrane cholesterol within the canalicular membrane prevented the cytotoxic effects of external bile acids. Therefore, the resident lipid transporting proteins must function together, converging on the goal to maintain homeostasis [59]. 

The cohort of proteins embedded in the canalicular bilayer performs varying tasks, contributing to the maintenance of lipid levels within the cell, membrane, and bile. Functioning as an endogenous sterol and phytosterol transporter, ABCG5/G8 is involved in a perpetual push and pull with its sterol importing counterpart, Niemann-Pick C1-Like 1 (NPC1L1). As substrates bind, NPC1L1′s conformation changes, resulting in the formation of the cholesterol delivering tunnel, in turn, allowing the substrates to navigate towards the outer leaflet of the membrane [61]. Multidrug resistant protein 3 (MDR3) also known as ABCB4, is a phospholipid translocase embedded in the canalicular membrane [62]. Although it actively flips inner leaflet phospholipids and sterols to the outer leaflet, there is evidence supporting its subsequent role in substrate secretion towards the bile [63,64,65]. Furthermore, MDR3 is found to be essential for the proper function of ABCG5/G8, which is strongly believed to be caused by its involvement in the formation of mixed micelles through phospholipid secretion in tandem with ABCB11′s bile salt secretion [66,67]. Mixed micelles are very charged, small aggregates of phospholipids, cholesterol and bile salts, and these micelles form the basis of currently known ABCG5/G8 acceptor particles [68,69]. Acceptor particles will then intake the exported lipids from ABCG5/G8; however, it is unknown if direct protein-particle interactions are necessary. Besides bile micelles, recent studies detected the presence of Niemann Pick C2 (NPC2), a lysosomal cholesterol transporting protein, in the bile [70]. Lack of NPC2 was shown to downregulate biliary cholesterol secretion in vivo, whereas using an NPC2-overexpressing mouse model, the ABCG5/G8 expression was positively regulated and the cholesterol secretion was restored. This suggests that although mixed micelles are considered the “primary” acceptor particle of sterols from ABCG5/G8, biliary NPC2 could speculatively play a “secondary” acceptor role by expediting the sterol transport from ABCG5/G8 to mixed micelles with its quick load and unload rates. Further studies remain needed to confirm direct interactions between the two proteins and between the NPC2 and mixed micelles. Anatomically, the bile feeds through the common bile duct which links to the upper small intestine known as the duodenum and trickles to the medially located jujenum. As ABCG5/G8 is highly expressed on the duodenum and jujenum brush border enterocytes [11], the presence of lipid rafts [71] and bile may provide a similar plasma membrane microenvironment to that on the bile ducts for ABCG5/G8 functions (Figure 3B).

### 4.2. ABCG1

The cholesterol-bound structures of ABCG1 show an inward-facing conformation. Two cholesterol molecules have been observed to be located in symmetrical alignment in the cholesterol binding site (Figure 2B). Using the numbering of a recent cryo-EM structure (PDB ID: 7R8D), the hydrophobic interactions between cholesterol and the binding site are formed through the residues of Phe455, Met459, and Leu463 from TM2 of one monomer and with Phe555, Pro558, Val 559 and Ile562 from TM5 of the other monomer (Figure 2B—right panel). Site-directed mutagenesis supports the essential roles of these residues in cholesterol transport [72]. In both of these cholesterol-bound structures of ABCG1, cholesterol shows similar binding interactions, although in one (PDB ID: 7FDV), cholesterol is located slightly higher [35,72]. 

In addition to cholesterol, other known lipid substrates like SM are also seen superficially on the cavity’s exterior. How the SM molecule enters the protein and gets transported remains elusive, although it may equally facilitate the recruitment and entry of cholesterol, possibly through an SM binding site in the inner leaflet cavity [73,74]. ABCG1 then promotes lipid translocation from the inner to outer leaflet and helps prevent the cytotoxic effects of excess cholesterol and sphingomyelin packing within the membrane, which would impact membrane fluidity and dynamics and cause membrane protein misfolding. ABCG1 thus appears to regulate its own function by dissipating the high cholesterol environment in which it is localized to [39], an environment shared by ABCG1, ABCA1, and Scavenger receptor- B1 (SR-B1), all participating in RCT (Figure 3D). ABCG1 has been long theorized to transport sterols to HDL molecules. Although, this topic remains contentious between two theories. The first argues that ABCG1 redistributes free cholesterol from the inner to outer leaflet of the membrane. HDL particles would then gather sterols by binding the membrane [48,75]. The other theory suggests ABCG1 effluxes ligands into extracellular pools, where the lipoproteins would intake them [76,77,78] (Figure 3C). HDL can bind membranes without the presence of protein intermediates [79], and at high enough concentrations, will remove outer leaflet cholesterol. Yet, at low lipoprotein concentrations, only extracellular cholesterol can be picked up [77]. These extracellular pools are exclusively caused by ABCG1 as ABCA1 is unable to perform this task, likely due to the temporary storage of ligands within its ECD [80] (Figure 3D). ABCA1 exports ligands to apoA-1, through direct and indirect ways. Around 10% of apoA-1 binds ABCA1 directly [81], while the remaining removed outer leaflet free cholesterol pools, whose formation could be the result of both ABCG1 and ABCA1, albeit this remains a debated topic [48,82,83,84] (Figure 3D).

## 5. Outlook

### 5.1. Challenge to Reveal Conformations of the Catalytic Cycle during the Sterol Transport

Structural analyses of ABCG1 and ABCG5/G8 agree on a number of enzymatic and genetic data and provide a wealth of molecular platforms to further investigate the structure-function relationship of ABCG sterol transporters by additional biochemical and biophysical approaches. Currently, several models exist in the inward-facing conformation, whether in the presence or absence of antibodies or nucleotide ligands [35,41,72]; yet there is only one nucleotide-bound outward-facing structure [25]. At the modest resolution of 3–4 Å, essentially all inward-facing models can be superimposed to the first ABCG5/G8 crystal structure. This suggests that, whether by X-ray crystallography or cryo-EM, there is still a huge gap and urgent need to capture intermediate conformations during the catalytic cycle. This will remain a major challenge in the coming decades, but technological advancement in either crystallography or electron microscopy will hold great promise. In addition, with the sophistication of molecular dynamics simulation, rigorous biophysical studies and computation characterizations, will address the dynamic aspects of sterol transporter mechanism.

### 5.2. Challenge to Determine Substrate Specificity and Selectivity on ABCG Sterol Transporters

Recent structures of ABCG1 or ABCG5/G8 revealed several bona fide sterol molecules. As cholesterol is part of lipid bilayers of cellular membranes, it may be tricky to differentiate substrate sterols and membrane structural sterols. Using computer-based molecular docking analysis, we observed differential cholesterol binding profiles on either transporters, suggesting sterol selectivity on ABCG1 and ABCG5/G8. In addition, ABCG1 appears to bind preferably cholesterol, suggesting substrate specificity among ABCG sterol transporters [41]. To support this hypothesis, it is necessary to exploit functional reconstitution of native and purified ABCG sterol transporters in their native membrane lipid environments and in the presence of a variety of sterol lipids and nucleotides, which is yet understudied. With the advancement of molecular dynamics simulation of large macromolecules, the existing structures and models of ABCG5/G8 and ABCG1 should promote interdisciplinary biochemical, biophysical and computational approaches to elucidate substrate specificity and sterol selectivity among ABCG sterol transporters.

## Figures and Tables

**Figure 1 ijms-24-00484-f001:**
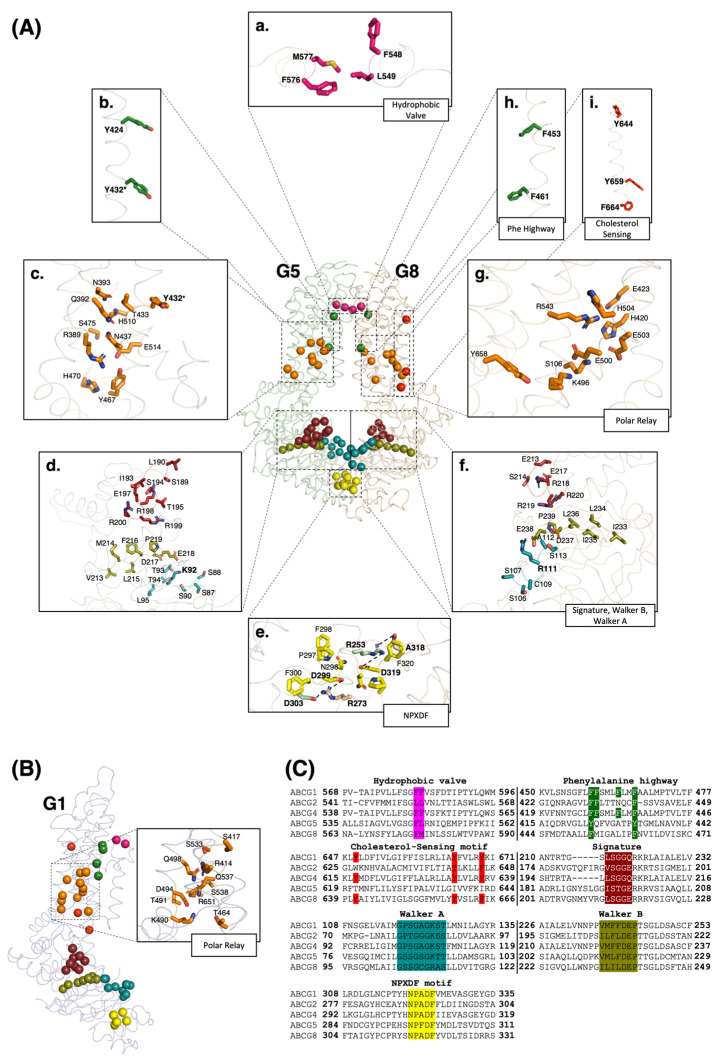
Motifs and elements in ABCG1 and ABCG5/G8. (**A**) Stick illustration of structural motifs shown on the crystal structure of ABCG5/G8 (PDB ID: 8CUB). (**a**–**i**) In the extracellular region, hydrophobic valve residues are shown in pink; in the TMDs, phenylalanine highway, cholesterol-sensing motif, and polar relay are represented in green, red, and orange, respectively; in the NBDs, signature, Walker A, Walker B, and NPXDF motifs are illustrated in brick red, deep teal, deep olive, and yellow, respectively. ***** The tyrosin residue labelled in the polar relay and phenylalanine highway is common in these two motifs. (**B**) The same conserved motifs with comparable colors in the structure of an ABCG1 half transporter. The *zoomed-in* picture shows the involved residues appear to form a polar relay network on the TMD of an ABCG1 monomer. These residues are highly conserved in mammals (see Appendix A). (**C**) The sequence alignment illustrates conserved residues of structural motifs shown in panels A and B in all ABCG subfamily members (colors are picked in accordance with the color code in panels (**A**,**B**)).

**Figure 2 ijms-24-00484-f002:**
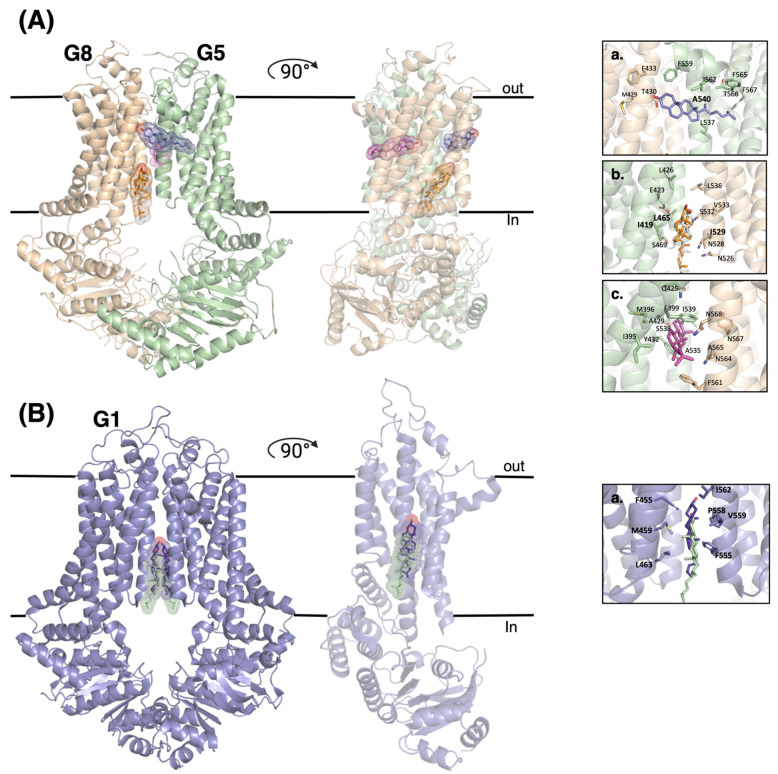
Cholesterol binding sites in the structures of ABCG1 and ABCG5/G8 proteins. (**A**) Comparison of cholesterol binding sites found on the determined structures of ABCG5/G8, as well as the involving residues. (**a**) This site (Site 3) was found in the crystal structure of ABCG5/G8 (PDB ID: 8CUB). (**b**,**c**). These binding sites (sites 1 and 2) were observed in the Cryo-EM structures of ABCG5/G8 (PDB ID: 7R8B and 7R8A, respectively). (**B**) Cartoon representation of ABCG1 in complex with two cholesterols forming a binding site. (**a**) Two cholesterol molecules colored in purple and green are from Cryo-EM structures of ABCG1, (PDB ID: 7FDV) and (PDB ID: 7R8D), respectively.

**Figure 3 ijms-24-00484-f003:**
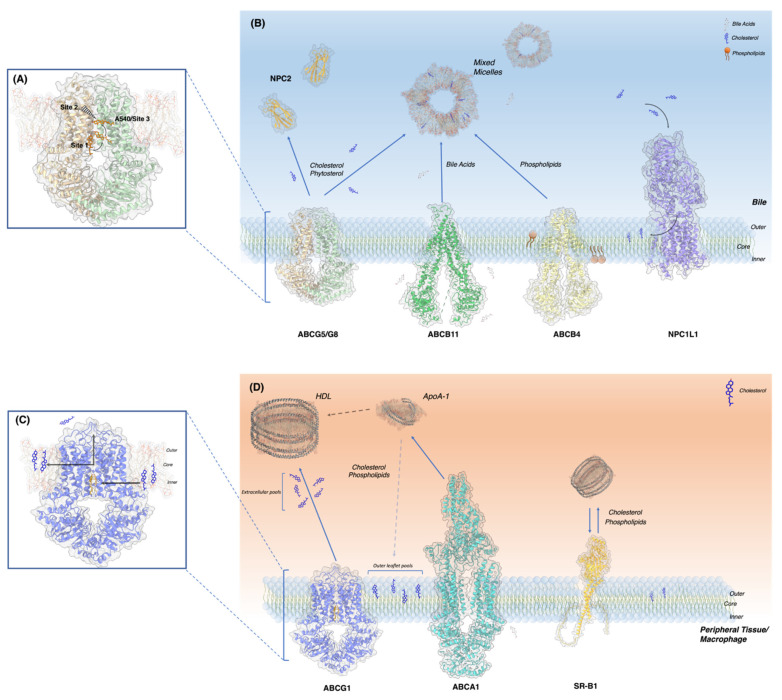
Molecular and cellular view of ABCG5/G8 and ABCG1 in their microenvironments. (**A**) Binding sites 1–3 of ABCG5/G8. Site 2 is on the back side of the protein. (**B**) Lipid and bile salt transporters of the canalicular membrane including their substrates and acceptor molecules, ABCG5/G8, ABCB11, ABCB4, NPC1L1 and NPC2 shown in cartoon. (PDB ID: 8CUB, 6LR0, 7NIV, 7DF8, 1NEP). (**C**) ABCG1 binding site and cholesterol trajectory towards protein and post efflux. (**D**) ABCG1 and lipid transporters expressed on macrophage: ABCA1, ABCG1 and SR-B1 (PDB ID: 7R8D, 5XYJ) (Alphafold).

## Data Availability

Not applicable.

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
