# Peer review of "Snapshots of ABCG1 and ABCG5/G8: A Sterol’s Journey to Cross the Cellular Membranes"

_ijms, 2022, doi:10.3390/ijms24010484_

Round 1

Reviewer 1 Report

Lee and colleagues summarized existing structures of ABCG1/G1 homodimer and ABCG5/G8 heterodimer on sterol transports across the cellular membrane, highlighting what is known and what is not. It is insightful to a reader who is not working on the field. However, I am struggling to see and understand what is in Figure 1 and what is described in the text. (1) Panel A (a-f) are too small for me to see what is in each subpanel. (2) How color codes in (C) related to (A) and (B). I do not see gold color which are one of the large groups in both (A) and (B) but not in (C). Sequence alignment in (C) should have sequence numbers in both ends on each line. (3) I cannot find hydrogen bond interactions of ABCG5(Arg253) with ABCG8(Asp319 and Asp323) described in the text, nor those of ABCG8(Arg273) with ABCG5(Asp299 and Asp303) in Figure 1A. Should residues described in the main text be labeled with bold and large fonts in Figure 1A so that the reader can easily spot them? (4) What do black bars stand for in (C), and is there a residue underneath it?

Two minor points:

Line 150. PH is not a good abbreviation for Phenylalanine highway. Is PheHW  better? Does the abbreviation need in the text?

Line 104: What are "two custeines"?

Author Response

See attached file for responses.

Reviewer 2 Report

In this review, Rezaei et al describes snapshots of ABCG1 and ABCG5/G8 : a sterol’s journey to cross the cellular membrane. Indeed, they have published a few months ago the structural analysis of cholesterol binding and sterol selectivity by ABCG5/G8. In this review, they focused their research on both ABCG1 and ABCG5/G8 and their implication in cholesterol homeostasis. This manuscript brings another synthetized view after the publication of the structure of both rules of ABCG1 and ABCG5/G8 in cholesterol homeostatis and transport, highlighted the importance of certain AA in these mechanisms.

Major revisions :

l174 : the authors should give more informations on the functional study [49].

Figure 2 : the authors should replace the a,b,c in their order of appearance in the text : a l192 // b l196 et c l 201. They should change the figure accordingly. Moreover, they should replace d by only inset since it corresponds to another part of the figure

The authors should include a reference on in vivo mouse biliary cholesterol transport (l220).

L253-ll255: the authors should reformulate the sentences and explain what they consider as acceptor particles.

Concerning NPC2, they should explain more in detail the phenomenon : is NCP2 secreted into the bile or does this protein help others ? how this protein interact with ABCG5/G8 ?

L259-260 : the authors have to add more informations on bile and link with the intestine.

L278-l282 : the authors should reformulate the sentences to help to the understanding on structures and their results

L330 : the authors have to reformulate the sentence starting by to support this hypothesis

Minor revisions :

l77 : to be expressed

l159 : through

l160 : recruits

l166 to l173 : at comparable sites // showed // and prevented its localization to cell surface

L175 : to be closed

L177 : G1 instead of G2

Legend figure 1 : lacks of color informations used in the alignment

L230 : SM ?

The authors should add references (l236-237)

L248: it actively flips

L261 : environment

L285 : it instead of sphingomyelin

L292 : what is RCT ?

L309: structural

L314 : nucleotide

L 321 : characterizations

L333: advancement of molecular dynamics

Author Response

See attached file for the responses.

Round 2

Reviewer 2 Report

Major revisions : 

l174 -- the authors have to give more details, describe how the efflux activity was impaired

l264-270 : the authors have to reformulate the sentence on the shipment between npc2 and the abcg proteins

l271-275: the authors should explain what is a working environment for a transporter and/or use other more appropriate words

l278-282 : the sentence even changed is still unclear. it lacks informations in which structure the cholesterol is located higher. the authors should add either figure or PDB ID. 

Moreover, in this paragraph, the amino acid residues have been changed. why ?

minor revisions :

in the legend of Figure 1 : tyrosine and phenylalanine must be written the right way

l243 : the word leaflets bring informations and should not be suppressed

l269 : interactions

l342: nucleotide 

Round 3

Reviewer 2 Report

by including modifications the manuscript into clarity and understanding.

only one sentence should be changed (part in italics) in line 276 : secondary acceptor role speculatively by expediting the sterol shipment. Indeed, sterols are not shiped. the authors should find another word for the possible function of NPC2. In addition the word speculatively should be placed between could and play. 

Author Response

Response to Reviewers’ Comments:

(New line numbers according to “Simple Markup” under “Track Changes”)

We thank the reviewer for the suggestion, and please find our point-to-point response below.

“only one sentence should be changed (part in italics) in line 276 : secondary acceptor role speculatively by expediting the sterol shipment. Indeed, sterols are not shipped. the authors should find another word for the possible function of NPC2. In addition the word speculatively should be placed between could and play.”

We have modified this sentence to improve clarity and understanding. (Lines 271-273)